# Infection with *Blastocystis* spp. and its association with enteric infections and environmental enteric dysfunction among slum-dwelling malnourished adults in Bangladesh

**Shah Mohammad Fahim** [1]*, **Md. Amran Gazi**[1], **Md. Mehedi Hasan**[1], **Md. Ashraful Alam** [1], **Subhasish Das**[1], **Mustafa Mahfuz**[1,2], **M. Masudur Rahman** [3], **Rashidul Haque** [4], **Shafiqul Alam Sarker** [1], **Tahmeed Ahmed**[1,5,6]

1 Nutrition and Clinical Services Division, International Centre for Diarrhoeal Disease Research, Bangladesh (icddr,b), Dhaka, Bangladesh, 2 Faculty of Medicine and Life Sciences, University of Tampere, Tampere, Finland, 3 Department of Gastroenterology, Sheikh Russel National Gastroliver Institute and Hospital, Dhaka, Bangladesh, 4 Infectous Diseases Division, International Centre for Diarrhoeal Disease Research, Bangladesh (icddr,b), Dhaka, Bangladesh, 5 Department of Global Health, University of Washington, Seattle, Washington, United States of America, 6 Department of Public Health Nutrition, James P Grant School of Public Health, BRAC University, Dhaka, Bangladesh

* mohammad.fahim@icddrb.org

**Data Availability Statement:** All relevant data are within the manuscript and its Supporting Information files.

## Abstract

### Background

*Blastocystis* spp. (*Blastocystis*) is a widely distributed gastrointestinal protist frequently reported in countries with tropical and sub-tropical climate. We sought to determine the factors associated with *Blastocystis* infection and investigate its role on biomarkers of intestinal health among slum-dwelling malnourished adults in Bangladesh.

### Methodology

Total 524 malnourished adults with a body mass index ≤18.5 kg/m$^2$ were included in this analysis. Presence of *Blastocystis* in feces was evaluated by TaqMan Array Card assays.

### Principal findings

*Blastocystis* was tested positive in 78.6% of the participants. Prevalence of infection with atypical strains of enteropathogenic *Escherichia coli* (aEPEC) (56% vs. 38%, p<0.001), and *Trichuris trichiura* (28% vs. 15%, p-value = 0.02) was significantly greater in adults with *Blastocystis*, while *Giardia intestinalis* was significantly lower (8% vs. 14%, p-value = 0.04) in *Blastocystis* positive adults. Malnourished adults who were living in households with high crowding index (aOR = 2.18; 95% CI = 1.11, 4.65; p-value = 0.03), and infected with aEPEC (aOR = 2.14; 95% CI = 1.35, 3.44; p-value = 0.001) and *Trichuris trichiura* (aOR = 1.97; 95% CI = 1.08, 3.77; p = 0.03) were more likely to be infected with *Blastocystis*. A significant negative relationship was observed between *Blastocystis* and fecal concentrations of alpha-1

**Funding:** This work was supported by the Bill and Melinda Gates Foundation under its Global Health Program [project investment ID is OPP1136751; https://www.gatesfoundation.org/How-We-Work/Quick-Links/GrantsDatabase/Grants/2015/11/OPP1136751] to TA. The funder had no role in study design, data collection and analysis, decision to publish, or preparation of the manuscript.

**Competing interests:** The authors have declared that no competing interests exist.

antitrypsin (β = -0.1; 95% CI = -1.7, -0.1; p-value<0.001) and Reg1B (β = -3.6; 95% CI = -6.9, -3.0; p-value = 0.03).

## Conclusions

The study findings suggest that the presence of *Blastocystis* in human intestine influences gut health and may have potential pathogenic role in presence of other pathogens.

## Author summary

Infection with *Blastocystis*, a neglected enteric pathogen, is frequently reported in tropical and sub-tropical countries. However, the epidemiology, pathogenicity and associated outcomes of *Blastocystis* in relation to enteric infections and Environmental Enteric Dysfunction (EED) remain inconclusive. In this study, the authors investigated the factors associated with *Blastocystis* infection and examined its association with enteric infections as well as fecal biomarkers of environmental enteric dysfunction among slum-dwelling malnourished adults in Bangladesh. The study findings exhibited that nearly 80% of the participants were infected with *Blastocystis* demonstrating a positive association with crowded living conditions. The authors observed a positive association of *Blastocystis* with atypical strain of Enteropathogenic *Escherichia coli* and *Trichuris trichiura*. The study results also corroborate that infection with *Blastocystis* had significant negative association with alpha-1 antitrypsin and Reg1B concentrations measured in the stool samples of the study participants. The findings of this study would help to reveal the pathogenic potential of *Blastocystis* and determine its role in contributing to altered gut health as well as EED in malnourished adults living in resource limited environments.

## Introduction

*Blastocystis* spp. (*Blastocystis*) is a widely distributed gastrointestinal protist frequently reported in the countries with tropical and sub-tropical climate [1,2]. The pathogen is transmitted by faeco-oral route and commonly found in human feces [3]. Although the prevalence varies from 30% to 76% in developing countries, the organism is assumed to be non-pathogenic [4]. Earlier studies suggest a significant contribution of *Blastocystis* in shaping the gut ecosystem in human [4,5]. Moreover, this single cell eukaryote (SCE) was found to be associated with a better immune status and richness of gut microbiota [4]. An association of this organism has been reported with the markers of healthy gut microbiota [6]. Then again, it was found to be associated with decrease of fecal microbiota protective bacteria, for instance, *Bifidobacterium sp.* and *Faecalibacterium prausnitzii* [7]. Evidence also suggests that there may be a potential link of this organism with Irritable Bowel Syndrome (IBS) [8]. Data from several studies reported a positive association between infection with *Blastocystis* and IBS [9,10]. *Blastocystis* was also found to be associated with inflammatory bowel disease [11,12]. A higher prevalence of *Blastocystis* was observed in patients with colorectal carcinoma with a significant positive association between the infection and colorectal cancer [13,14]. It has also been found to be responsible for acute gastroenteritis [15]. Besides, it was reported to cause alterations in intestinal barrier function and impair the gut permeability [9,16].

Most people who carry the organism have no signs or symptoms, but it can also be found in people who have diarrhea and other digestive problems [9,17]. Most importantly, *Blastocystis*

often appears with other pathogenic organisms, particularly in immunosuppressed individuals [18,19]. Therefore, *Blastocystis* can be an opportunistic organism and may have adverse role on gut health in presence of other pathogens. However, evidence pertaining to its pathogenicity in human subjects remains speculative and inconclusive [17,20]. There is no convincing evidence to ascertain whether *Blastocystis* causes disease or it is beneficial for microbial diversity in our gut. Moreover, there is no data regarding the contribution of *Blastocystis* in the pathogenesis of Environmental Enteric Dysfunction (EED) among adults living in resource limited settings. EED is another asymptomatic entity known to cause disturbances in normal intestinal function and nutrient absorption, which ultimately leads to impaired physical development [21,22]. Although EED is caused by sustained exposure to fecal pathogens similar to *Blastocystis*, to date there is no definitive answer regarding the relationship between EED and infection with *Blastocystis*. Furthermore, the epidemiology, pathogenicity and associated outcomes of *Blastocystis* remain ambiguous. Therefore, we have designed this study with an aim to determine the factors associated with *Blastocystis* infection in malnourished adults living in a slum. We also sought to investigate the role of *Blastocystis* infection on the biomarkers of EED among the same population.

## Methodology

### Ethics statement

This study was approved by the Institutional Review Board of the International Center for Diarrheal Disease Research, Bangladesh (icddr,b). Informed written consent was obtained from all the participants prior to enrolment.

### Study design, site and population

We used data from the Bangladesh Environmental Enteric Dysfunction (BEED) study for this analysis. The BEED study was a community-based nutrition intervention study where adults with a body mass index (BMI) less than 18.5 kg/m$^2$ were enrolled between March 2016 to September 2019. The study was conducted in a slum-setting in Dhaka, the capital of Bangladesh. The age range of the participants was 18–45 years. Severe anemia, tuberculosis, psychiatric disorders and presence of any chronic diseases were the exclusion criteria. Pregnant and lactating women were also excluded from the study. Socio-demographic information and anthropometry data were collected by trained field staff at enrollment. Blood and non-diarrheal stool samples were also collected in the baseline for laboratory assays. The objectives of the BEED study include investigating the effectiveness of nutrition intervention in improving the nutritional status of the participants, examining the role of enteric pathogens in the pathogenesis of EED and malnutrition, and development of a histological scoring system to identify EED and validate the score against non-invasive biomarkers of EED. We have highlighted the screening and recruitment of study participants in a flowchart in supporting files (Fig A in S1 File). However, the detailed methodology of the BEED study has been published earlier [23].

### Laboratory assays

Laboratory assays were done at icddr,b. Plasma biomarkers including C-reactive protein (CRP) (Immundiagnostik, Bensheim, Germany), alpha-1-acid glycoprotein (AGP) (Alpco, Salem, NH), sCD14 (R&D systems Inc, Minnesota, USA), retinol binding protein-4 (RBP4) (R&D systems Inc, Minnesota, USA), ferritin (ORGENTEC Diagnostika GmbH, 55129 Mainz, Germany) and low-density lipoprotein receptor-related protein-1 (LRP1) (BIOMATIK, Wilmington, USA) were analyzed using commercial ELISA kits. Plasma Zinc was

measured by atomic absorption spectrometry. HemoCue Hb 201 System was used to measure hemoglobin level. Fecal biomarkers of EED, for example, Myeloperoxidase (MPO) (Alpco, Salem, New Hampshire), Neopterin (NEO) (GenWay Biotech, San Diego, California), Alpha-1 anti-trypsin (A1AT) (Biovendo Chandler, North Carolina), Reg1B (TechLab, Blacksburg, Virginia), and Calprotectin (BUHLMANN fCAL, Schonenbuch, Switzerland) were assessed by ELISA. *Helicobacter pylori* stool antigen was also detected using ELISA (Amplified IDEIA Hp StAR, OXOID Limited, Hampshire, United Kingdom). In this study, presence of enteropathogens including *Blastocystis* spp. and other intestinal parasites were detected in the stool samples by a quantitative PCR assay using the TaqMan Array Cards (TAC) [24]. It is a 384-well singleplex, probe-based real-time PCR assay for detecting pathogens in the fecal samples. The analytic cut-off for this pathogen was a Ct-value (cycle threshold) of $\geq$35. Thus, a Ct value <35 was considered positive for the pathogen [25,26]. We have analyzed single fecal sample per participant collected at enrolment to detect the enteropathogen and the assay was done once for each sample. Although routinely three fecal samples are examined to identify intestinal parasites, we used single specimen considering the analytical performance of TAC assay compared to other laboratory methods. The TAC assay can detect multiple targets even with a very small copy numbers (as low as 0.1 to 500 copies per 1 μl of reaction mixture), while conventional microscopic method may miss the detection of intestinal pathogens if the sample processing is not done properly. Moreover, the average sensitivity and specificity of TAC assay are 85% and 77%, respectively when compared with conventional methods such as microscopic examination, culture, and immunoassay. The sensitivity and specificity of TAC analysis were recorded even higher (98% and 96%, respectively) when compared with laboratory-developed PCR-Luminex assays [24].

## Upper gastro-intestinal endoscopy and histopathology

We conducted a sub-analysis on participants (n = 50) who underwent upper gastro-intestinal endoscopy. The procedure was performed by gastroenterologists with expertise in managing patients with digestive disorders. We obtained biopsy tissues from second part of the duodenum and an expert histopathologist examined the tissues to determine the pathology. Among the participants who underwent endoscopy, 42 were tested positive for *Blastocystis*.

## Variables used in this analysis

We considered infection with *Blastocystis* as the outcome variable in order to determine the factors associated with this infection. It was a binary categorical variable that has been categorized based on the presence of *Blastocystis* in the stool samples. The covariates such as age, sex, BMI, education, employment status, treatment of drinking water, source of drinking water, source of cooking water, hand washing practice after toilet, hand washing practice before cooking, improved sanitation, use of toilet paper, separate space for kitchen, chicken or ducks at households, monthly family income, and household crowding index [e.g. Low ($\leq$4 people sleep in a room), High (>4 people sleep in a room)] were considered as independent variables. Anemia, iron deficiency, and zinc deficiency were also considered. For the linear regression analyses, fecal biomarkers EED, for instance, MPO, NEO, Calprotectin, Reg1B and A1AT were the outcome variables.

## Statistical analyses

Demographic characteristics were expressed as mean with standard deviation (SD) or frequency with proportion estimate, as appropriate. To assess the differences between groups, t-test or Mann-Whitney test was performed for numeric data, and chi-square or fisher's exact

test was done for categorical variables. Multivariable logistic regression analysis was used to identify the factors associated with *Blastocystis* infection. The covariates were included in the model if they demonstrate a p-value <0.2 in the bivariate analyses. Additionally, we adjusted the model with age and sex of the participants. We then performed multivariable linear regression analysis to investigate the association of *Blastocystis* with biomarkers of intestinal health. Herein, fecal biomarkers of gut enteropathy, for instance, MPO, NEO, Calprotectin, Reg1B and A1AT were the outcome variables. The biomarkers were normalized prior to inclusion in the model. All the individual models were adjusted by the covariates if bivariate analyses showed a p-value <0.2. Statistical analyses were performed using R version 3.5.3. We considered a probability of <0.05 as statistically significant.

## Results

A total of 524 adults with a BMI less than 18.5 kg/m$^2$ were included in this analysis. Of them, 72.7% were female and 4.6% were severely underweight. The mean (±SD) age of the participants was 23.8 (±6.9) years. The detailed socio-demographic information is described in Table 1.

### Prevalence of *Blastocystis*

Overall, 78.6% malnourished adults were tested positive for *Blastocystis* (Fig 1). The prevalence was higher among female participants compared to male adults (80.1% vs. 74.8%, p-value = 0.22), and in those who were mildly underweight compared to adults with moderate and severe underweight (79.9% vs. 77.3 vs. 70.8%, p-value = 0.51). We also observed an increased prevalence among the participants who kept chicken or ducks at home (84.2% vs. 78.4%, p-value = 0.78) and who did not have a separate space for kitchen (84.9% vs. 77.7%, p-value = 0.85). However, none of these differences were statistically different (p-value>0.05).

**Table 1. Descriptive characteristics of the study participants.**

| Variables, n (%) | Without *Blastocystis* (n = 112) | With *Blastocystis* (n = 412) | Overall (n = 524) |
|---|---|---|---|
| Age in years, mean (SD) | 24.4 (6.6) | 23.7 (6.9) | 23.8 (6.9) |
| Sex | | | |
| Male | 36 (32.1%) | 107 (26.0%) | 143 (27.3%) |
| Female | 76 (67.9%) | 305 (74.0%) | 381 (72.7%) |
| Body mass index, mean (SD) | 17.2 (0.9) | 17.3 (0.8) | 17.3 (0.8) |
| Nutritional status | | | |
| Mild underweight | 67 (59.8%) | 266 (64.6%) | 333 (63.5%) |
| Moderate underweight | 38 (33.9%) | 129 (31.3%) | 167 (31.9%) |
| Severe underweight | 7 (6.3%) | 17 (4.1%) | 24 (4.6%) |
| Monthly income in BDT, mean (SD) | 16621 (8549.3) | 15957 (9751.2) | 16099.0 (9503.2) |
| Water treatment (Yes) | 70 (62.5%) | 250 (60.1%) | 320 (61.1%) |
| Improved sanitation (Yes) | 14 (12.5%) | 65 (15.8%) | 79 (15.1%) |
| Use of toilet paper (Yes) | 24 (21.4%) | 91 (22.1%) | 115 (22.0%) |
| Always wash hand after toilet (Yes) | 85 (75.9%) | 325 (78.9%) | 410 (78.2%) |
| Always wash hand before cooking or preparing foods (Yes) | 13 (11.7%) | 75 (18.6%) | 88 (16.8%) |
| Crowding index | | | |
| Low (≤ 4 people sleep in a room) | 101 (90.2%) | 336 (81.6%) | 437 (83.4%) |
| High (> 4 people sleep in a room) | 11 (9.8%) | 76 (18.4%) | 87 (16.6%) |
| Separate space for kitchen (Yes) | 102 (91.1%) | 356 (86.4%) | 458 (87.4%) |
| Kept chickens/ducks at home (Yes) | 6 (5.5%) | 32 (7.8%) | 38 (7.3%) |

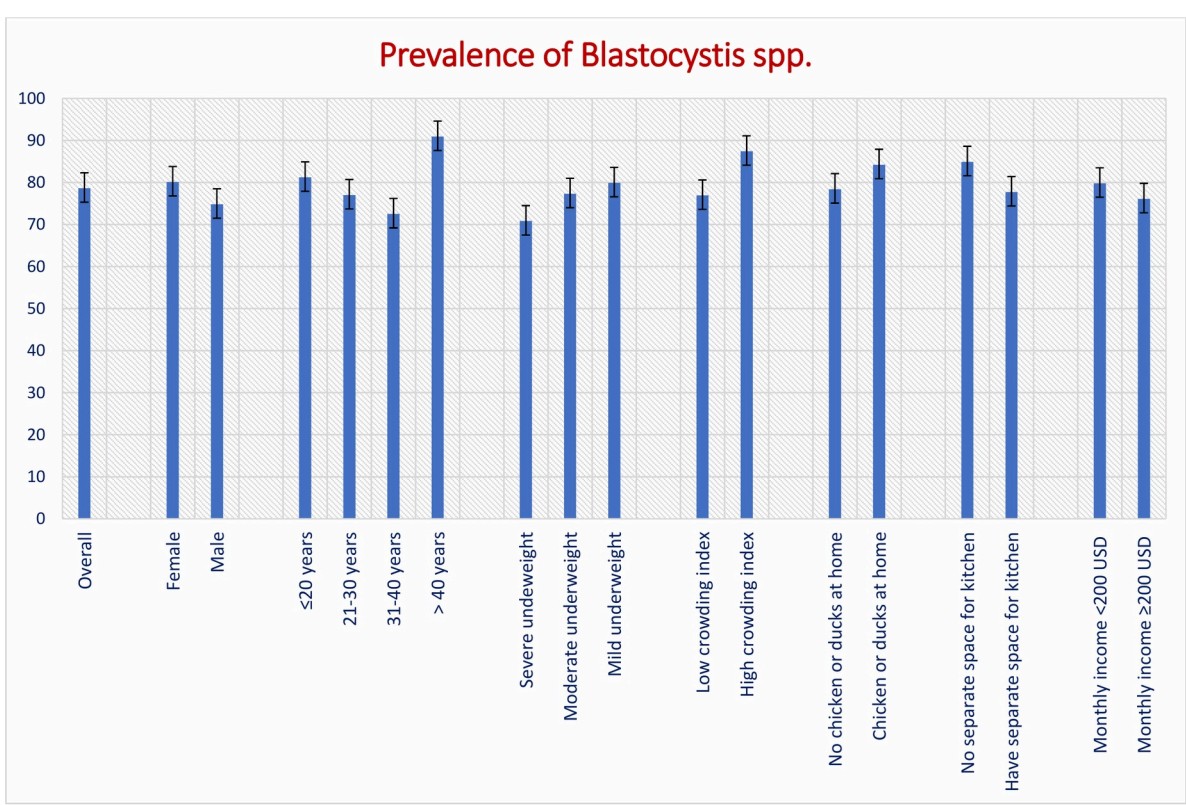

**Fig 1. Prevalence of *Blastocystis* in slum-dwelling malnourished adults in Bangladesh.**

Only the difference based on household crowding index demonstrated a statistical significance. Malnourished adults living in households with high crowding index had significantly higher prevalence of *Blastocystis* compared to their counterparts (87.4% vs. 76.9%, p-value = 0.03).

### Micronutrient deficiencies and Seasonal trend in the prevalence of *Blastocystis*

The prevalence of anemia, iron deficiency, and zinc deficiency were 37.4%, 14.9% and 23.1%, respectively among the malnourished adults recruited in this study (Fig B in S1 File). There was no difference in the prevalence of these micronutrient ailments between the adults with *Blastocystis* and those who were tested negative for the pathogen. We observed a higher prevalence of *Blastocystis* during winter season that extends from December to February (Fig C in S1 File). The prevalence went down during summer and this downward trend continued till monsoon. Then again, the positivity of *Blastocystis* started to rise during post-monsoon months. However, the difference in seasonal trend of prevalence was statistically insignificant (p-value = 0.06).

### Enteropathogens in malnourished adults with and without *Blastocystis*

Overall, 91.5% participants had more than one pathogen in their stool samples, mostly contributed by the bacterial agents. Only 1.5% participants had no bacteria in their fecal samples, while more than one bacterial pathogen were detected in 88.3% of the participants. The proportion of virus and parasites detected in stool samples was null in 82.4% and 65.7% of the

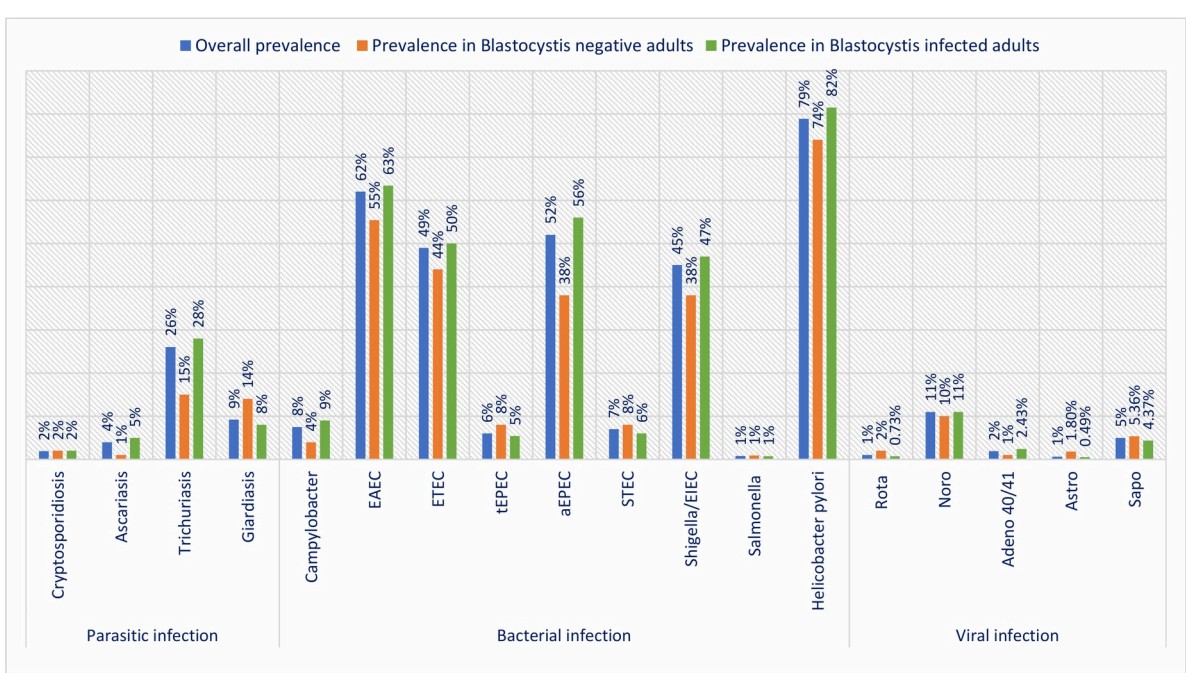

**Fig 2. Prevalence of asymptomatic enteric infections according to infection with *Blastocystis* infection among malnourished adults in Bangladesh.**

participants, respectively (Fig D in S1 File). Fig 2 illustrates the prevalence of asymptomatic enteric infections according to infection with *Blastocystis* infection among the malnourished adults. Prevalence of infection with aEPEC (56% vs. 38%, p-value<0.001), and *Trichuris trichiura* (28% vs. 15%, p-value = 0.02) were estimated significantly greater in adults with *Blastocystis* infection. On the other hand, *Giardia intestinalis* was detected significantly lower in adults with *Blastocystis* compared to those who were tested negative for the pathogen (8% vs. 14%, p-value = 0.04). An elevated prevalence of *Helicobacter pylori* (82% vs. 74%, p-value = 0.08), *Shigella*/EIEC (47% vs. 38%, p-value = 0.09) and EAEC (63% vs. 55%, p-value = 0.14) was also observed among the adults with *Blastocystis*, although the differences were not statistically significant. No significant difference was reported in the prevalence of viral organisms between the groups with and without infection.

### Distribution of plasma and fecal biomarkers among adults with *Blastocystis*

There was no difference in distribution of CRP, sCD14 and RBP4 among the adults with and without *Blastocystis* (Fig E in S1 File). However, the differences of AGP and LRP1 was statistically significant. We observed a lower plasma concentration of AGP in adults tested positive for *Blastocystis* (p-value = 0.03), while LRP1 measured in plasma samples was found elevated in adults with *Blastocystis* (p-value = 0.003). The concentrations of MPO, calprotectin, Reg1B and A1AT were lower in the stool samples of the *Blastocystis* infected malnourished adults. However, only the differences for Reg1B (p-value = 0.001) and A1AT (p-value<0.001) concentrations between infected and non-infected participants were found statistically significant (Fig F in S1 File). We observed that concentrations of A1AT, MPO, NEO, and calprotectin were elevated in *Blastocystis*-positive malnourished adults who were severely underweight (defined by a BMI less than 16.0 kg/m$^2$) compared to mild and moderately underweight adults of this study (Fig G in S1 File). Likewise, a number of fecal biomarkers were measured high in

severely underweight adults infected with several enteric pathogens, for instance, *Shigella*, *Helicobacter pylori*, STEC, ETEC, EAEC, aEPEC, tEPEC, *Giardia intestinalis*, *Trichuris trichiura*, Norovirus and Sapovirus (Fig G in S1 File).

## Factors associated with *Blastocystis* in malnourished adults

Multivariable logistic regression analysis demonstrated that malnourished adults living in crowded households had higher odds of being infected by *Blastocystis* (aOR = 2.18; 95% CI = 1.11, 4.65; p-value = 0.03) after controlling for potential confounders (Table 2). The likelihood of infection with *Blastocystis* was found significantly lower in monsoon season (aOR = 0.43; 95% CI = 0.22, 0.83; p-value = 0.01) compared to winter. Likewise, *Blastocystis* infection was lower among the *Giardia*-infected adults in bivariate analysis, although it became insignificant (aOR = 0.55; 95% CI = 0.28, 1.11; p-value = 0.09) in multivariable model. However, the odds of infection with *Blastocystis* were significantly greater among the participants

**Table 2. Factors associated with *Blastocystis* infection in malnourished adults[1].**

| Variables | OR (95% CI) | p-value | aOR (95% CI) | p-value |
|---|---|---|---|---|
| Age, years | 0.99 (0.96, 1.02) | 0.33 | 0.98 (0.95, 1.01) | 0.23 |
| Sex (Ref: Female) | | | | |
| Male | 0.74 (0.47, 1.17) | 0.19 | 0.83 (0.49, 1.40) | 0.47 |
| Body Mass Index, kg/m$^2$ | 1.11 (0.87, 1.43) | 0.40 | | |
| Anemia | 1.23 (0.76, 2.02) | 0.42 | | |
| Iron deficiency | 1.43 (0.78, 2.81) | 0.27 | | |
| Zinc deficiency | 0.93 (0.58, 1.54) | 0.78 | | |
| Water treatment | 0.93 (0.60, 1.42) | 0.73 | | |
| Improved Sanitation | 1.07 (0.92, 1.26) | 0.39 | | |
| Use of toilet paper | 1.04 (0.63, 1.76) | 0.88 | | |
| Always wash hand after toilet | 1.19 (0.72, 1.93) | 0.50 | | |
| Always wash hand before cooking/preparing food | 1.72 (0.95, 3.37) | 0.09 | 1.74 (0.91, 3.52) | 0.11 |
| Separate space for kitchen | 0.62 (0.29, 1.22) | 0.19 | 0.50 (0.22, 1.04) | 0.08 |
| Kept chickens or ducks at home | 1.47 (0.64, 3.99) | 0.40 | | |
| Crowding index (Ref: Low) | | | | |
| High (> 4 people sleep in a room) | 2.08 (1.10, 4.27) | **0.03** | 2.18 (1.11, 4.65) | **0.03** |
| Season (Ref: Winter) | | | | |
| Summer | 0.78 (0.38, 1.59) | 0.49 | 0.88 (0.90, 1.95) | 0.76 |
| Monsoon | 0.46 (0.24, 0.85) | **0.02** | 0.43 (0.22, 0.83) | **0.01** |
| Autumn | 0.78 (0.41, 1.43) | 0.42 | 0.85 (0.44, 1.62) | 0.63 |
| *Helicobacter pylori* | 1.59 (0.96, 2.58) | 0.06 | 1.46 (0.85, 2.47) | 0.16 |
| *Shigella*/EIEC | 1.48 (0.97, 2.29) | 0.07 | 1.14 (0.70, 1.85) | 0.61 |
| EAEC | 1.39 (0.91, 2.13) | 0.12 | 1.16 (0.73, 1.85) | 0.52 |
| aEPEC | 2.11 (1.38, 3.25) | <0.001 | 2.14 (1.35, 3.44) | **0.001** |
| tEPEC | 0.65 (0.30, 1.52) | 0.29 | | |
| *Trichuris trichiura* | 1.97 (1.15, 3.55) | **0.02** | 1.97 (1.08, 3.77) | **0.03** |
| *Giardia intestinalis* | 0.51 (0.27, 0.98) | **0.04** | 0.55 (0.28, 1.11) | 0.09 |

[1]Multivariable logistic regression model was adopted and adjusted for the variables with p-values <0.20 in the univariate logistic regression analysis.

OR, odds ration; aOR, adjusted odds ratio; CI, confidence interval; EIEC, Enteroinvasive *Escherichia coli*; EAEC, Enteroaggregative *Escherichia coli;* aEPEC, atypical strains of enteropathogenic *Escherichia coli;* tEPEC, typical enteropathogenic *Escherichia coli*.

infected with aEPEC (aOR = 2.14; 95% CI = 1.35, 3.44; p-value = 0.001) and *Trichuris trichiura* (aOR = 1.97; 95% CI = 1.08, 3.77; p-value = 0.03).

## Association of *Blastocystis* spp. with fecal biomarkers of gut enteropathy

Fig 3 shows the relationship between *Blastocystis* and fecal biomarkers–A1AT and Reg1B. *Blastocystis* infection was significantly associated with both the biomarkers in multivariable linear regression analyses. *Blastocystis* infection was inversely associated with the fecal concentrations of A1AT (β = -0.67; 95% CI = -0.98, -0.36; p-value<0.001) after adjusting for potential confounders, and the association was statistically different. Similarly, a statistically significant negative association was observed between *Blastocystis* infection and fecal concentrations of REG1B (β = -0.66; 95% CI = -1.06, -0.26; p-value = 0.001) in adjusted model. However, *Blastocystis* was not found to be associated with MPO, NEO and calprotectin.

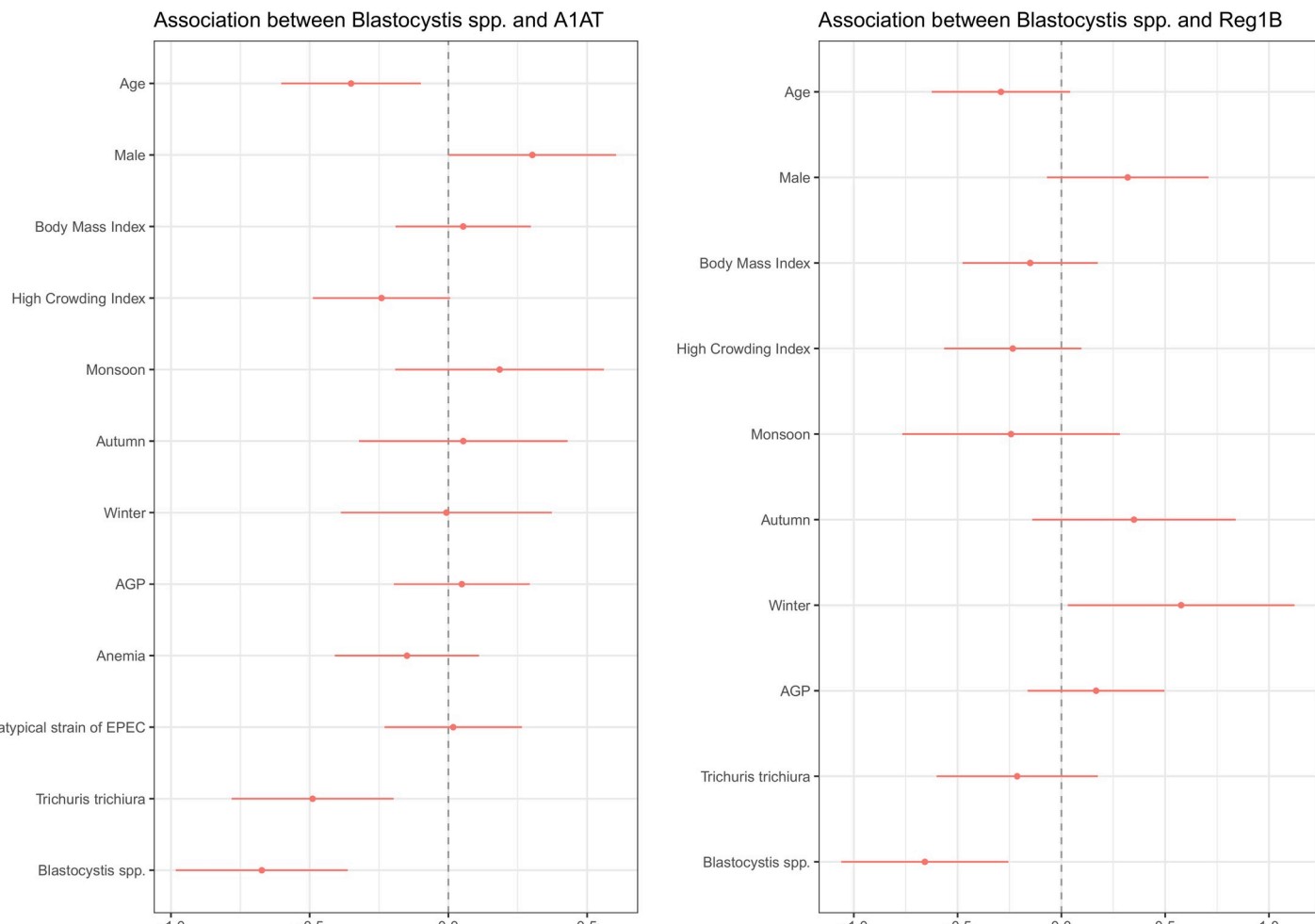

**Fig 3. Forest plots showing the relationship between *Blastocystis* and fecal biomarkers of Environmental Enteric Dysfunction in malnourished adults.** The illustration on left side demonstrates the association between *Blastocystis* and A1AT, while the photograph in right side shows the relationship between *Blastocystis* and Reg1B. The results were obtained by multivariable linear regression analyses after adjusting for the variables with p-values <0.20 in the univariate linear regression analysis. Herein, the fecal biomarkers, for instance, A1AT and Reg1B, were the dependent variables. A1AT, alpha-1 anti-trypsin; Reg1B, Regenerating Family Member 1 Beta; AGP, alpha-1-acid glycoprotein; EPEC, enteropathogenic *Escherichia coli*.

## Differences in findings of upper gastro-intestinal endoscopy and histopathology

We did not observe any significant differences in endoscopic findings among adults with and without *Blastocystis* (Fig H in S1 File). Endoscopic pathologies, for instance, gastritis, ulcer and gastric erosion, were diagnosed more in malnourished adults with *Blastocystis*. Histological findings such as villous atrophy, crypt hyperplasia and increased intraepithelial lymphocytes were greater in participants without *Blastocystis*. However, all of these differences were statistically insignificant. We also observed a statistically insignificant higher proportion of campylobacter-like organism (CLO) positivity in study participants tested negative for *Blastocystis*. However, 81% of the participants with *B. hominis* were also tested positive for CLO, the test for detection of *Helicobacter pylori* from biopsy tissues.

## Discussion

To the best of our knowledge, this is the first study to delineate the prevalence and associated outcome of *Blastocystis* among the undernourished adults, particularly in a low-income setting. Our study results revealed that more than three-fourth of the malnourished adults had *Blastocystis* in their stool samples, with an increased prevalence among the participants living in crowded households. We also reported that adults with low BMI living in crowded households had higher odds of being infected by *Blastocystis*. These observations support the historical findings on abundance of *Blastocystis* in developing countries owing to poor personal hygiene, unsanitary living conditions, consumption of contaminated food and water, and exposure to animals [9,27,28]. Moreover, the prevalence of *Blastocystis* reported in this study is consistent with previous studies done in developing world [4,29,30]. *Blastocystis* is transmitted through fecal-oral route. Earlier reports documented that unhygienic environmental conditions as well as crowded households of developing countries increases the transmission of enteric pathogens [22]. Nonetheless, malnourished individuals demonstrate a higher risk of infection with enteropathogens [31]. It is, perhaps, due to compromised immune status which increases the susceptibility to enteric infections [32]. As we observed a positive association between *Blastocystis* and high crowding index, and all of our study participants were malnourished, it may explain the higher prevalence of *Blastocystis* in this study. The observation of this analysis in relation to seasonal influences is counterintuitive compared to prior works [33–35]. The infection was more prevalent during winter followed by post-monsoon. The probability of *Blastocystis*-detection was lower during monsoon–commonly referred to rainy season in Bangladesh. Perhaps, geographical distribution and outdoor activities may explain the results related to seasonal variation [36]. Outdoor activities remain restricted during rainy season which may further limit the source of infections. Moreover, the dry and dusty climate during winter may enhance the transmissibility of the infective form of this enteric pathogen [34,36].

Our findings did not replicate prior reports which found *Blastocystis* to be associated with impaired barrier function and altered gut permeability [9,16]. Our results displayed lower concentrations of A1AT and Reg1B in the feces of *Blastocystis*-positive adults. We also noticed a significant negative relationship of *Blastocystis* with A1AT and Reg1B. A1AT is a marker of intestinal protein loss and indicates increased gut permeability [37,38]. Whereas Reg1B is a marker of epithelial regeneration and healing [22]. Prior evidence revealed association of such biomarkers with enteric infections [22,39]. Sustained exposure to fecal pathogens possibly alters the gut health and results in elevation of fecal biomarkers of impaired intestinal functions [39]. However, the inverse association observed in this study goes in line with earlier works suggesting *Blastocystis* as a beneficial organism for gut health [6,40]. We have noted

lower plasma AGP and increased LRP1 levels in *Blastocystis*-infected malnourished adults. These findings also reinforce the idea of *Blastocystis* being an advantageous gut commensal.

On the other hand, the findings on relationship between *Blastocystis* and a number of fecal pathogens exhibited an ambiguous nature of this SCE. We did not observe any significant relationship of *Blastocystis* with ascariasis, hookworm infection, infection with *Entamoeba histolytica*, and *Cryptosporidium* species, albeit these are considered as common parasitic infections in countries similar to Bangladesh [41]. However, the prevalence of ascariasis, hookworm infection, *Entamoeba histolytica*, and *Cryptosporidium* species were inconsiderable (4%, 0.8%, 1%, and 2%, respectively) among the study participants which may explain the insignificant results of co-invasion between *Blastocystis* and these parasites. Conversely, we observed an increased prevalence of certain enteric pathogens including aEPEC and *Trichuris trichiura* in study participants with *Blastocystis*. This finding confirms the previous reports of *Blastocystis* being an opportunistic organism [19,42]. It was also reported earlier that *Blastocystis* can be found in conjunction with other intestinal pathogens including *Trichuris trichiura* [43]. A recent Australian study revealed an inverse relationship between *Blastocystis* and height-for-age z-scores [44]. Perhaps, *Blastocystis* causes damage to intestinal mucosa in presence of other pathogens and leads to impaired nutrient absorption and growth faltering. It is also possible that pathogenicity of *Blastocystis* is related to specific subtypes or strain of the pathogen [40]. In addition, compromised immune status and disruption of normal gut flora may play a vital role to exert the pathogenic potential of *Blastocystis* [43]. However, further exploration is required to unveil the association between *Blastocystis* subtypes and disease-causing enteric pathogens.

Our sub-analysis on participants who underwent upper gastro-intestinal endoscopy did not demonstrate any differences. However, we observed 81% of the biopsy samples collected from the malnourished adults with *Blastocystis* were tested positive for *Helicobacter pylori*. This is consistent with previous work demonstrating increased frequency of *Helicobacter pylori* in individuals with *Blastocystis* positivity [45].

This study has several limitations. First, we only included adults with low BMI in this analysis. The absence of a healthy control group limits the strength of our findings. Second, the disproportion in female: male ratio of the participants. This study was a community-based intervention trial where it was mandatory to visit the study site in every morning for two consecutive months in order to receive the nutrition intervention. Most of the male participants became ineligible as they informed that they may not stay in the study area for next two months, and may migrate to another place. In addition, a substantial number of eligible males were workers or wage-earners who refused to miss the work for an hour to receive the nutrition intervention. Hence, a higher number of female participants who were mostly homemakers or part-time employees were enrolled in this study. Third, *Blastocystis* is a genetically diverse protozoan and the pathogenic potential highly depends on its pan-genome [6]. Hence, detection at the strain/sub-type level would allow us to better understand the pathogenicity of this organism. Moreover, sub-type analysis would help us to determine the prevalence of *Blastocystis* subtypes in malnourished adults. Fourth, the absence of another recommended method (e.g. microscopic examination or PCR) to verify or confirm the positivity of *Blastocystis* in the stool samples is another important limitation of this study. Fifth, the use of single specimen per participant for parasitological examination while it is recommended to examine three fecal samples from each individual to identify intestinal parasites. Finally, we could not include the clinical features of the participants that may have provide us further insights regarding the clinical consequences. However, the strength of this study includes use of quantitative TAC PCR assays for detection of *Blastocystis* and other enteric pathogens. TAC assay is considered to be a highly sensitive molecular diagnostic method for determination of a wide

range of enteropathogens [25]. The inclusion of a broad array of socio-demographic variables enabled us to control the effect of potential confounders. Moreover, the study of a large number of participants and a detailed survey are the strengths of the study.

In conclusion, the study findings suggest that presence of *Blastocystis* in human intestine influences gut health and may have potential pathogenic role in presence of other organisms. The pathogen seems to stifle certain enteric pathogens and ameliorate the intestinal health of malnourished adults. On the other hand, it potentiates the presence of aEPEC and *Trichuris trichiura*–the organisms known for causing gut infection and altered intestinal health. Therefore, we recommend further evaluation of the relationship of this ambiguous pathogen with harmful enteropathogens as well as biomarkers of EED, with special consideration on detection of subtypes of *Blastocystis*.

## Supporting information

**S1 STROBE Checklist. Checklist of items that should be included in reports of observational studies.**
(DOC)

**S1 File. Supplementary figures.**
(DOC)

**S2 File. Supplementary tables.**
(DOCX)

## Acknowledgments

The authors express thanks to the participants of the study as well as to the field and laboratory staffs at icddr,b for their valuable contributions. icddr,b is also grateful to the Government of Bangladesh, Canada, Sweden and the UK for providing unrestricted support.

## Author Contributions

**Conceptualization:** Shah Mohammad Fahim, Tahmeed Ahmed.

**Data curation:** Shah Mohammad Fahim, Md. Ashraful Alam.

**Formal analysis:** Shah Mohammad Fahim.

**Funding acquisition:** Tahmeed Ahmed.

**Investigation:** Shah Mohammad Fahim, Md. Amran Gazi, Md. Mehedi Hasan, Rashidul Haque.

**Methodology:** Shah Mohammad Fahim, Md. Amran Gazi, Md. Ashraful Alam, Subhasish Das, Mustafa Mahfuz, Shafiqul Alam Sarker, Tahmeed Ahmed.

**Project administration:** Shah Mohammad Fahim, Subhasish Das, Mustafa Mahfuz, Tahmeed Ahmed.

**Resources:** Mustafa Mahfuz, Rashidul Haque, Tahmeed Ahmed.

**Supervision:** M. Masudur Rahman, Rashidul Haque, Shafiqul Alam Sarker, Tahmeed Ahmed.

**Writing – original draft:** Shah Mohammad Fahim.

**Writing – review & editing:** Md. Amran Gazi, Md. Mehedi Hasan, M. Masudur Rahman, Rashidul Haque, Shafiqul Alam Sarker, Tahmeed Ahmed.

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
