## [Decision Letter · Decision Letter 0]

26 Mar 2021

Dear Dr Fahim,

Thank you very much for submitting your manuscript "Infection with Blastocystis hominis and associated outcomes in relation to enteric infections and environmental enteric dysfunction among slum-dwelling malnourished adults in Bangladesh" for consideration at PLOS Neglected Tropical Diseases. As with all papers reviewed by the journal, your manuscript was reviewed by members of the editorial board and by several independent reviewers. In light of the reviews (below this email), we would like to invite the resubmission of a significantly-revised version that takes into account the reviewers' comments. 

All 3 reviewers have indicated that the current study is of interest to the scientific community. The authors should take note of the concerns and suggestions of the reviewers, with particular attention to the comments of Reviewer #2, including the need for some form of validation for the high % of Blastocystis infection among the cohort examined. The reviewers have also indicated more details on the methods, such as the collection procedure (e.g. how many samples collected) should be provided. Please note that the term Blastocystis hominis is no longer used and the accepted terminology is Blastocystis spp. The lack of subtype information should be discussed as a limitation of the study.

We cannot make any decision about publication until we have seen the revised manuscript and your response to the reviewers' comments. Your revised manuscript is also likely to be sent to reviewers for further evaluation.

Sincerely,

Kevin SW Tan

Associate Editor

Shan Lv

Deputy Editor

All 3 reviewers have indicated that the current study is of interest to the scientific community. The authors should take note of the concerns and suggestions of the reviewers, with particular attention to the comments of Reviewer #2, including the need for some form of validation for the high % of Blastocystis infection among the cohort examined. The reviewers have also indicated more details on the methods, such as the collection procedure (e.g. how many samples collected) should be provided. Please note that the term Blastocystis hominis is no longer used and the accepted terminology is Blastocystis spp. The lack of subtype information should be discussed as a limitation of the study.

Reviewer's Responses to Questions

**Key Review Criteria Required for Acceptance?**

**Methods**

-Are the objectives of the study clearly articulated with a clear testable hypothesis stated?

-Is the study design appropriate to address the stated objectives?

-Is the population clearly described and appropriate for the hypothesis being tested?

-Is the sample size sufficient to ensure adequate power to address the hypothesis being tested?

-Were correct statistical analysis used to support conclusions?

-Are there concerns about ethical or regulatory requirements being met?

Reviewer #1: The study acceptable for publication.

Study design is appropriate for the aim.

Authors mentioned limitations of this study. Lacking of the healthy control group and subtype analysis of Balstocystis are the main limitations for this study.

Some suggestions were written for correction in the main text and supp2.

Yes the sample size is sufficient to ensure adequate power to address the hypothesis being tested

Yes, correct statistical analysis was used to support conclusions

Yes, there concerns about ethical or regulatory requirements are being met

Reviewer #2: - It is a routine practice in parasitological examinations to examine three faecal samples for each patient (such samples being taken at appropriate intervals). It is not specified how many samples per one patient were analyzed and how many times each trial was repeated. This information should be provided.

- The percentage of infected patients is very high. Therefore, the results for positive samples should be confirmed/verified using another method recommended by CDC (e.g. fecal smears, PCR).

Reviewer #3: Minor Revision.

Partly related to STROBE Items 6,9, and 13. While the inclusion and exclusion criteria is specified, the paper refers to the BEED study protocol in BMJ Open for details. The reference is a protocol that describes the recruitment process, but does not describe the outcome of recruitment. The paper can refer to a publication with the outcome of recruitment, or if this is the first from that protocol, perhaps provide more details in the Results section. Undernutrition in Bangladesh between females and males show a higher prevalence of malnutrition among females (23% vs 20%; https://globalnutritionreport.org/resources/nutrition-profiles/asia/southern-asia/bangladesh/). However, the study has 73% females. What steps in the recruitment process resulted in this ratio?

**Results**

-Does the analysis presented match the analysis plan?

-Are the results clearly and completely presented?

-Are the figures (Tables, Images) of sufficient quality for clarity?

Reviewer #1: Yes, the analysis was presented match the analysis plan.

Yes, the results are clearly and completely presented

Yes the figures (Tables, Images) of sufficient are quality for clarity

Reviewer #2: - Molecular method used in these study -Quantitative PCR assay using the TaqMan Array Cards- allows obtaining quantitative data. Such data should be presented and correlated with the indications of the studied biomarkers. The ms needs to be supplemented with this.

- Results with regard to association of Blastocystis hominis with fecal biomarkers should be summarized in a table.

Reviewer #3: Minor Revision. Related to the comments in Methods. It might be useful to describe the flow of recruitment to consent and reasons for refusal to better understand the ratio of females to males in this study.

**Conclusions**

-Are the conclusions supported by the data presented?

-Are the limitations of analysis clearly described?

-Do the authors discuss how these data can be helpful to advance our understanding of the topic under study?

-Is public health relevance addressed?

Reviewer #1: Yes, the conclusions are supported by the data presented

Yes, the limitations of analysis are clearly described

Yes, the authors discuss how these data can be helpful to advance our understanding of the topic under study.

Reviewer #2: - The most common parasitic infections in Bangladesh are Ascariasis, Hookworm, Trichuriasis, and Cryptosporidium. However, in the ms it is not explained why only co-invasion with T. trichiura was taken into account.

- The high results of Blastocystis hominis vs. study participants and associated factors should be discussed.

Reviewer #3: Minor revisions. Related to Strobe Item 19 and points above. Should the female : male ratio in this study be considered a limitation? How would this affect internal and external validity .

The title uses the phrase "associated outcomes" which tend to suggest to me, that the outcomes are a result of B. hominis infection. The study design is cross-sectional and given that is the case, any association of B. hominis infection and fecal biomarkers and other pathogens is just that - an association. I think the paper is best served by stating this as assocations clearly.

**Editorial and Data Presentation Modifications?**

Reviewer #1: The only modifications needed are minor. So my decision is minor revision. Authors should be made all of them.

Reviewer #2: .

Reviewer #3: Consider the best use of graphs and tables as some of the figures can easily be represented by a table. 

For instance, figure 1 shows the prevalence of B. hominis according to different sub-groups, but do not show the confidence intervals that would have demonstrated that the differences are not significant and reinforced the text. That information could be contained in a table or a graph of a point estimate bounded by 95% confidence intervals.

Figure 2 shows the prevalence of infection of each pathogen defined by the TAC assay. This could also have been visualized as a table or heatmap which could also show the extent of co-infection. This information would also reinforce the text.

I am particularly interested in the distribution of positive results of all enteropathogens and enteric biomarkers as a categorical or quantitative variable sorted by BMI. By choosing to show aggregate data, you lose the value of the powerful methods you have used. Hence the suggestion of a heatmap above. I think this might in fact be very useful to explore the how infection / inflammation (whether due to B hominis or not) is distributed across the range of malnourished adults, if only to generate new hypotheses.

Lines 204-207 and 227-230 may be repeating information on REG1B. If it is, perhaps, just place that information in one place.

**Summary and General Comments**

Reviewer #1: I think the publication of this study, which is aimed at explaining the pathogenesis of the blastocytsis protist parasite and includes a large number of samples, will make a great contribution to the researcher.

It will benefit both in terms of creating new hypotheses and in terms of evaluating the researchers ' own work with the data of this study.

However, as the authors noted, the absence of a particularly healthy control group and the absence of subtype analysis of Blastocystis, as well as the absence of microscopic examination of stool samples, are an important limitations. It will be useful for the TaqMan array method used to indicate the pathogens it is investigating and present these identified pathogens in a table. A standard investigation of all enteric pathogens and the study of a large number of patients and a detailed survey are the strengths of the study.

Reviewer #2: This ms presents the data on Blastocystis prevalence in malnourished adult patients in Bangladesh. The authors investigated the relationship of B. hominis with the occurrence of intestinal biomarkers. The data are interesting and can be used to develop new diagnostic methods. However, I have some doubts about the parasitological research.

Analysis of results is clear but the parasitological analysis is insufficient. They should be completed.

Reviewer #3: This is a relevant paper and addresses the possible associations of B. hominis infection with other pathogens and host factors.

PLOS authors have the option to publish the peer review history of their article (what does this mean?). If published, this will include your full peer review and any attached files.

Reviewer #1: No

Reviewer #2: No

Reviewer #3: No
---

## [Decision Letter · Decision Letter 1]

23 Jun 2021

Dear Dr Fahim,

Thank you very much for submitting your manuscript "Infection with Blastocystis spp. and its association with enteric infections and environmental enteric dysfunction among slum-dwelling malnourished adults in Bangladesh" for consideration at PLOS Neglected Tropical Diseases. As with all papers reviewed by the journal, your manuscript was reviewed by members of the editorial board and by several independent reviewers. The reviewers appreciated the attention to an important topic. Based on the reviews, we are likely to accept this manuscript for publication, providing that you modify the manuscript according to the review recommendations. 

The reviewers have completed their assessment of the revised manuscript. Reviewers #1 and #3 are supportive of accepting the current version. However, Reviewer #2 has indicated 2 minor edits to the manuscript before acceptance. These are to provide details and correlations of the TaqMan qPCR data with the assessed biomarkers, and to provide details on lab identification of Ascariasis, Hookworm infestation, Entamoeba histolytica, and Cryptosporidium species, which is currently missing in the ms. Both these aspects can be included in the supplementary section of the manuscript.

Sincerely,

Kevin SW Tan

Associate Editor

Shan Lv

Deputy Editor

The reviewers have completed their assessment of the revised manuscript. Reviewers #1 and #3 are supportive of accepting the current version. However, Reviewer #2 has indicated 2 minor edits to the manuscript before acceptance. These are to provide details and correlations of the TaqMan qPCR data with the assessed biomarkers, and to provide details on lab identification of Ascariasis, Hookworm infestation, Entamoeba histolytica, and Cryptosporidium species, which is currently missing in the ms. Both these aspects can be included in the supplementary section of the manuscript.

Reviewer's Responses to Questions

**Key Review Criteria Required for Acceptance?**

**Methods**

-Are the objectives of the study clearly articulated with a clear testable hypothesis stated?

-Is the study design appropriate to address the stated objectives?

-Is the population clearly described and appropriate for the hypothesis being tested?

-Is the sample size sufficient to ensure adequate power to address the hypothesis being tested?

-Were correct statistical analysis used to support conclusions?

-Are there concerns about ethical or regulatory requirements being met?

Reviewer #1: Yes

Reviewer #2: It is a routine practice in parasitological examinations to examine three faecal samples for each patient (such samples being taken at appropriate intervals). The present research has only been done from one trial. The authors suggest that the method used in this study has a high detection rate. However, they do not take into account the fact that dispersive forms of parasites can be excreted periodically and taking only one sample for analysis significantly reduces the detection of, for example, parasite eggs.

Molecular method used in the study -Quantitative PCR assay using the TaqMan Array Cards - allows obtaining quantitative data. Such data should be presented and correlated with the indications of the studied biomarkers. The ms needs to be supplemented with this.

Reviewer #3: (No Response)

**Results**

-Does the analysis presented match the analysis plan?

-Are the results clearly and completely presented?

-Are the figures (Tables, Images) of sufficient quality for clarity?

Reviewer #1: Yes

Reviewer #2: (No Response)

Reviewer #3: (No Response)

**Conclusions**

-Are the conclusions supported by the data presented?

-Are the limitations of analysis clearly described?

-Do the authors discuss how these data can be helpful to advance our understanding of the topic under study?

-Is public health relevance addressed?

Reviewer #1: Yes

Reviewer #2: The discussion in lines 311-314 ["However, the prevalence of Ascariasis, Hookworm infestation, Entamoeba histolytica, and Cryptosporidium species were inconsiderable (4%, 0%, 1%, and 2%, respectively) among the study participants which may explain the insignificant results of co-invasion between Blastocystis and these parasites."] refers to results that are not described in the relevant sections of the paper, i.e.: materials and methods and the results. It is not known how these species of parasites were detected and why these studies were not described in Methods and Results.

Reviewer #3: (No Response)

**Editorial and Data Presentation Modifications?**

Reviewer #1: Revised manuscript Line 311 "Ascariasis, Hookworm infestation" should be changed as "ascariasis, hookworm infection". 

After his correction was made there is no need reviewing again by me.

Reviewer #2: (No Response)

Reviewer #3: (No Response)

**Summary and General Comments**

Reviewer #1: Dear Editor 

Thank you very much for reviewing invitation again for this manuscript.

I checked all my suggestion and other reviewew's comments. Authors tried to make all of corrections according to the reviewer's comments.

In this revised form, the manuscirpt has become better understood and scientifically enriched.

My decision is "accept" of this manuscript for publication.

Kind regards

Reviewer #2: This ms presents the data on Blastocystis prevalence in malnourished adult patients in Bangladesh. The authors investigated the relationship of B. hominis with the occurrence of intestinal biomarkers. The data are interesting and can be used to develop new diagnostic methods. However, I have some doubts about the parasitological research. 

The analysis of the results is clear but the parasitological analysis is still insufficient.

Reviewer #3: (No Response)

PLOS authors have the option to publish the peer review history of their article (what does this mean?). If published, this will include your full peer review and any attached files.

Reviewer #1: No

Reviewer #2: No

Reviewer #3: No

Figure Files:

Data Requirements:

Reproducibility:

References

---

## [Editor Report · Decision Letter 2]

26 Jul 2021

Dear Dr Fahim,

We are pleased to inform you that your manuscript 'Infection with Blastocystis spp. and its association with enteric infections and environmental enteric dysfunction among slum-dwelling malnourished adults in Bangladesh' has been provisionally accepted for publication in PLOS Neglected Tropical Diseases.

Best regards,

Kevin SW Tan

Associate Editor

Shan Lv

Deputy Editor

---

## [Editor Report · Acceptance letter]

13 Aug 2021

Dear Dr. Fahim,

We are delighted to inform you that your manuscript, "Infection with Blastocystis spp. and its association with enteric infections and environmental enteric dysfunction among slum-dwelling malnourished adults in Bangladesh," has been formally accepted for publication in PLOS Neglected Tropical Diseases.

Best regards,

Shaden Kamhawi

co-Editor-in-Chief

Paul Brindley

co-Editor-in-Chief
